# Cell-Type Apoptosis in Lung during SARS-CoV-2 Infection

**DOI:** 10.3390/pathogens10050509

**Published:** 2021-04-23

**Authors:** Yakun Liu, Tania M. Garron, Qing Chang, Zhengchen Su, Changcheng Zhou, Yuan Qiu, Eric C. Gong, Junying Zheng, Y. Whitney Yin, Thomas Ksiazek, Trevor Brasel, Yang Jin, Paul Boor, Jason E. Comer, Bin Gong

**Affiliations:** 1Department of Pathology, University of Texas Medical Branch, Galveston, TX 77555, USA; yakliu@utmb.edu (Y.L.); qichang@utmb.edu (Q.C.); zhsu@utmb.edu (Z.S.); chazhou@utmb.edu (C.Z.); yuqiu@utmb.edu (Y.Q.); ericgong128@gmail.com (E.C.G.); tgksiaze@utmb.edu (T.K.); 2Department of Microbiology and Immunology, University of Texas Medical Branch, Galveston, TX 77555, USA; tmgarron@utmb.edu (T.M.G.); trbrasel@utmb.edu (T.B.); 3Department of Neuroscience and Cell Biology, University of Texas Medical Branch, Galveston, TX 77555, USA; juzheng@utmb.edu; 4Department of Biochemistry and Molecular Biology, University of Texas Medical Branch, Galveston, TX 77555, USA; ywyin@utmb.edu; 5Division of Pulmonary and Critical Care Medicine, Department of Medicine, Boston University Medical Campus, Boston, MA 02118, USA; yjin1@bu.edu

**Keywords:** SARS-CoV-2, apoptosis, lung, human, non-human primate, co-culture, endothelial cell, epithelial cell, TUNEL assay

## Abstract

The SARS-CoV-2 pandemic has inspired renewed interest in understanding the fundamental pathology of acute respiratory distress syndrome (ARDS) following infection. However, the pathogenesis of ARDS following SRAS-CoV-2 infection remains largely unknown. In the present study, we examined apoptosis in postmortem lung sections from COVID-19 patients and in lung tissues from a non-human primate model of SARS-CoV-2 infection, in a cell-type manner, including type 1 and 2 alveolar cells and vascular endothelial cells (ECs), macrophages, and T cells. Multiple-target immunofluorescence assays and Western blotting suggest both intrinsic and extrinsic apoptotic pathways are activated during SARS-CoV-2 infection. Furthermore, we observed that SARS-CoV-2 fails to induce apoptosis in human bronchial epithelial cells (i.e., BEAS2B cells) and primary human umbilical vein endothelial cells (HUVECs), which are refractory to SARS-CoV-2 infection. However, infection of co-cultured Vero cells and HUVECs or Vero cells and BEAS2B cells with SARS-CoV-2 induced apoptosis in both Vero cells and HUVECs/BEAS2B cells but did not alter the permissiveness of HUVECs or BEAS2B cells to the virus. Post-exposure treatment of the co-culture of Vero cells and HUVECs with a novel non-cyclic nucleotide small molecule EPAC1-specific activator reduced apoptosis in HUVECs. These findings may help to delineate a novel insight into the pathogenesis of ARDS following SARS-CoV-2 infection.

## 1. Introduction

Three zoonotic coronaviruses (CoV) that belong to the genus *Batacoronavirus* within Coronaviridae have caused deadly pneumonia in humans since 2003: severe acute respiratory syndrome coronavirus (SARS-CoV), Middle East respiratory syndrome (MERS) coronavirus, and SARS-CoV-2 [1]. The latter is associated with the ongoing global outbreak of COVID-19 pneumonia [2,3]. The pandemic caused by SARS-CoV-2 has inspired renewed interest in understanding the fundamental pathological changes observed in acute respiratory distress syndrome (ARDS) because fatal COVID-19 cases are most commonly linked to respiratory failure with the clinical and pathologic features of ARDS [4,5]. Lung injury following SARS-CoV-2 infection has been proposed to result from endothelial and epithelial cell injury causing inflammation, an overwhelming cytokine response, and cell death [6,7,8,9,10,11,12]. Furthermore, evidence has shown that endothelial damage might play a role in multisystem inflammatory syndrome, leading to multiple organ failure in severe COVID-19 cases [6,13,14]. While a role for cell surface angiotensin converting enzyme 2 (ACE2) receptor as a target is proposed [15], the underlying mechanisms remain largely unclear.

The pathologic alteration known as diffuse alveolar damage (DAD) in endothelial and epithelial cells is a critical feature of acute lung injury in ARDS [16]. Bronchial epithelial cells, alveolar type I and II cells, and vascular endothelial cells (ECs) are proposed targets during the early stage of COVID-19 [11,17,18,19,20]. Apoptosis is the most extensively investigated form of cell death in the context of viral infections and has been detected in bronchial and lung epithelial cells during the initial exudative phase (day 2–4 post infection) of SARS-CoV-2 infection in Syrian hamsters [21] and humanized ACE2 transgenic mice [22]. By contrast, alterations in alveolar ECs are usually subtle, although endotheliopathy, including endotheliitis, endothelial coagulopathy, and apoptosis spur the fatal phase of ARDS. Given its exposure to aggressive inflammatory signals from the host in response to SARS-CoV infection, ECs are activated, and endothelial dysfunction occurs, shifting the vascular equilibrium toward inflammation marked by tissue edema and a procoagulant state. Clinic pathological evidence from fatal COVID-19 cases directly supports findings of endotheliitis, endothelial coagulopathy, and microthrombosis [17]. Similar endotheliopathy was documented in a non-human primate (NHP) model of SARS-CoV-2 infection during the initial exudative phase [18]. However, endothelial apoptosis was reported based on hematoxylin and eosin (H&E) staining in postmortem lung sections from COVID-19 patients who had complicated preexisting underlying diseases [6]. Evidence of endothelial apoptosis remains lacking in NHP [1,7,18,23,24] hamster [25,26,27,28,29] and mouse [15,30,31] models of SARS-CoV-2 infections. Furthermore, compared with lung epithelial cells, the EC is a relatively non-permissive cell to CoV infections in vitro [32]. Therefore, whether SARS-CoV-2 infection can induce a cell-type apoptosis in lung and whether apoptosis plays a regulatory role in the pathogenesis of SARS-CoV-2 infection remain to be elucidated.

In the present study, we observed apoptosis in postmortem lung sections from COVID-19 patients and lung tissues of NHPs in a model of SARS-CoV-2 infection, in a cell-specific manner, including type 1 and 2 alveolar cells, ECs, macrophages, and T cells. A multiple-target immunofluorescence (IF) staining and Western blotting suggest both intrinsic and extrinsic apoptotic pathways are activated during SARS-CoV-2 infection. Furthermore, we observed that SARS-CoV-2 fails to induce apoptosis in human primary human umbilical vein EC (HUVEC) and bronchial epithelial cell BEAS2B, which are refractory to SARS-CoV-2 infection. However, infection of the co-culture of Vero cells and HUVECs or Vero cells and BEAS2B cells with SARS-CoV-2 induced apoptosis in both Vero cells and HUVECs/BEAS2B cells but did not alter the permissiveness of HUVEC or BEAS2B cells to the virus. Given that the exchange protein directly activated by cAMP (EPAC) plays various roles in regulating cell apoptosis in different cells [33,34], we further found that post-exposure treatment of the co-culture of Vero cells and HUVECs with an EPAC1-specific activator (ESA) ameliorated apoptosis in HUVECs. These finding may help to delineate a novel insight into the pathogenesis of ARDS following SARS-CoV-2 infection.

## 2. Results

### 2.1. Similar Pulmonary Pathology Was Observed in Humans and Non-Human Primates Following SARS-CoV-2 Infection

COVID-19 pneumonia is a heterogeneous disease, which is primarily marked by tracheobronchitis, diffuse alveolar damage, and vascular injury [4,5]. DAD is considered the histological hallmark for the acute phase of ARDS. DAD is characterized by an acute phase with edema, hyaline membrane formation, and inflammation, followed by an organizing phase with alveolar septal fibrosis and type II pneumocyte hyperplasia. In especially long-standing and severe DAD during ARDS, extensive fibrin accumulates in alveolar spaces in greater amounts than seen with typical hyaline membranes. Intra-alveolar fibrin deposition in acute fibrinous and organizing pneumonia is known as “fibrin balls” [35].

First, H&E staining was performed on postmortem lung sections from five COVID-19 patients with different underlying diseases (Appendix A). Similar to what has been reported [17], lymphocytic pneumonia and extensive vasculopathy were detected in all cases (Appendix A) (Figure 1).

Next, we conducted a retrospective study of lung tissues from four NHPs on day 21 post-infection (p.i.) with SARS-CoV-2. Lymphocytic endotheliitis, perivasculitis, microthrombosis, and pneumonia with thickened septae were detected in all cases (Figure 2), a finding that has been reported by others [1,7,18,23]. Sporadic signals of SARS-CoV-2 viral antigens were detected in NHP lung tissue on day 21 p.i. (Appendix A).

Data from these retrospective pathology examinations supported our further investigations of apoptosis involving different cell-types.

### 2.2. Extensive Apoptotic Signals Were Detected in the Lung Tissues of Humans and NHPs Following SARS-CoV-2 Infections

To examine whether SARS-CoV-2 infection triggers apoptosis in human lungs, we performed the TUNEL assay on postmortem lung sections from five COVID-19 cases, all with different underlying disease. Compared with normal human lungs, we observed marked apoptotic signals in alveolar and vascular areas in lungs from four cases (Appendix A) (Figure 3).

Next, we conducted a retrospective study of lung tissue from four NHPs on day 21 p.i. with SARS-CoV-2. Compared with control NHP lung tissue, the TUNEL assay revealed remarkable apoptotic signals in all NHP SARS-CoV-2 infection cases (Figure 4). In addition, DNA fragmentation, one of the key features of apoptosis [36], was visualized using the DNA ladder assay in all NHP lung tissue samples, compared with the uninfected control (Appendix A).

These data demonstrated that SARS-CoV-2 infection induces extensive apoptosis in the lungs of humans and NHPs.

### 2.3. Cell Type Identification of Apoptosis in NHP Lung Following SARS-CoV-2 Infection

Histologically, SARS-CoV-2 infection-induced pulmonary pathology is characterized by epithelial and endothelial cell injury [6,17], as demonstrated in Figure 1 and Figure 2. Growing evidence shows that cell-type apoptosis contributes to ARDS pathogenesis [16]. Taking advantage of an established assay coupling in situ TUNEL and IF labeling, we analyzed cell-specific apoptosis in NHP lung tissue on day 21 p.i. with SARS-CoV-2. Analysis of confocal microscopy on TUNEL and IF double-labeling using antibodies against CD31 (an EC-specific marker [37]), caveolin (an alveolar type 1 cell marker [38]), and surfactant protein C (SPC) (an alveolar type 2 cell specific marker [38]) demonstrated apoptotic signals were predominantly visualized in ECs (Figure 4 and Figure 5), alveolar type 1 cells (Figure 6A–C), and alveolar type 2 cells (Figure 6E–G), respectively, in lung tissue of infected animals, compared with the uninfected control. Moreover, using antibodies against CD68 (a macrophage marker) and CD3 (a T cell marker) [38], apoptotic signals were also detected in macrophages (Figure 6I–K) and T cells (Figure 6M–O) in the lung tissue of infected animals, whereas control tissue lacked these apoptotic signals.

These data suggest that SARS-CoV-2 infection triggers programed cell death in different cells in lung tissue.

### 2.4. Both Intrinsic and Extrinsic Apoptotic Pathways Are Activated Following SARS-CoV-2 Infection

Apoptosis is initiated by intrinsic and extrinsic pathways. The intrinsic pathway is triggered by mitochondrial membrane permeabilization, leading to the release of cytochrome C and other proapoptotic factors into the cytoplasm [39,40]. The extrinsic pathway involves activation of the transmembrane death receptor family, including Fas and Fas ligand (FasL) [40]. Given that cytochrome C expression is required for TNFα-induced apoptosis in human cells [40], we examined the correlation between the TUNEL signal and the IF signal of cytochrome C or FasL. We observed TUNEL signals colocalized with enhanced cytochrome C (Figure 7A–C) and FasL (Figure 7E–G) signals, respectively, in vascular or alveolar areas. Relative frequency of cytochrome C- or FasL-positive apoptotic cells was assessed, suggesting both intrinsic and extrinsic pathways are activated following SASR-CoV-2 infections (Appendix A).

Vero cell infection is a well-established model for use in CoV infection-induced apoptosis mechanism studies [36,41]. The TUNEL assay demonstrated that SARS-CoV-2 infection induces apoptosis in Vero cells 72 h p.i. (Figure 8A–F). Furthermore, Western blotting was used to visualize cleavage of poly ADP-ribose polymerase (PARP) and caspase-9, an indication of apoptosis mediated by the intrinsic pathway [41], in Vero cells at 72 h p.i. with SARS-Cov-2, compared with the uninfected control (Figure 8G).

These data suggest SARS-CoV-2 infection-triggered apoptosis is mediated by both intrinsic and extrinsic signaling cascades.

### 2.5. SARS-CoV-2 Infection Triggers Apoptosis in Non-Permissive Cells

SARS-CoV-2 infects the host using the ACE2 receptor, which is abundantly expressed in the epithelia of lung and small intestine [15]. The expression of ACE2 on ECs suggests these cells are susceptible to virus entry. It has been shown that SARS-CoV-2 elements can be detected in ECs in fatal COVID-19 cases, along with vasculopathy [6,17]. However, HUVEC has been shown to be refractory to SARS-CoV [32]. In the present study, endothelial apoptosis was predominantly detected in both human and NHP lung tissues following SARS-CoV-2 infection (Figure 3 and Figure 4). Compared with Vero cells (Figure 8), we observed that HUVECs (Figure 8 and Figure 9C) and human bronchial epithelial BEAS2B cells (Figure 9B) are refractory to SARS-CoV-2 infection, as no viral antigen was detected, and no apoptosis was detected 72 h p.i. at an MOI of 0.1 of SARS-CoV-2.

To further explore the mechanism underlying in vivo SARS-CoV-2 infection-triggered apoptosis in relatively non-permissive ECs, we employed co-culture models of Vero cells and HUVECs or Vero cells and BEAS2B cells to mimic the in vivo cell-to-cell interfaces (Figure 9). First, co-culturing did not change the permissiveness to SARS-CoV-2 of these three types of cells. Compared with the mock controls, there was no viral antigen signal detected in HUVECs (Figure 9C) or BEAS2B cells (Figure 9B) of the co-culture when examined at 72 h p.i. with an MOI of 0.1 of SARS-CoV-2. As with the single-cell cultures, Vero cells in both co-cultures demonstrated apoptosis with positive staining for viral antigens (Appendix A). Interesting, an MOI of 0.1of SARS-CoV-2 in co-culture of Vero cells and HUVECs or Vero cells and BEAS2B cells induced apoptosis in both HUVECs (Figure 9C) and BEAS2B cells (Figure 9B), respectively, at 72 h p.i. without altering the permissiveness to the virus.

Collectively, these data suggest that cell apoptosis can be induced either directly or indirectly by SARS-CoV-2 infection in human cells.

### 2.6. Pharmacological Activators of EPAC1 Reduce Endothelial Apoptosis in Vero Cells and HUVECs co-Culture Following Infection with SARS-CoV-2

Classically, apoptosis is considered to be an antiviral strategy of the host to promote viral clearance [22,41]. However, growing evidence has revealed a positive correlation between apoptosis and the pathogenesis of virus-induced disease [6,11]. The cAMP-EPAC signaling axes play various roles in regulating cell apoptosis in different cells [33,34]. To date, two isoforms, EPAC1 and EPAC2, have been identified, and EPAC1 is the major isoform in ECs [42]. To evaluate the underlying mechanism(s) and identify potential intervention(s) to ameliorate SARS-CoV-2 infection-triggered apoptosis, the co-cultures of Vero cells and HUVECs were treated with the novel EPAC1-specific agonist ESA I942 (5µM) and EPAC-specific antagonist ESI NY0173 (5µM) [42] at 24 h p.i. Quantitative analysis of TUNEL signals suggested that post-exposure activation of EPAC with I942 attenuated apoptosis in ECs infected with SARS-CoV-2 (Figure 10) but not in the Vero cells (Appendix A), while post-exposure inactivation of EPAC enhanced apoptosis in ECs (Figure 10). Similar to the vehicle-only control group, no viral antigen was detected in ECs in the group treated with either I942 or NY0173 (Appendix A).

These data suggest that SARS-CoV-2 infection-induced endothelial apoptosis can potentially be controlled via cAMP-EPAC signaling pathway.

## 3. Discussion

Lung damage and multi-organ failure are features of COVID-19 pathogenesis [43]. Given that cytokine storm is associated with an aggressive inflammatory host response leading to an unfavorable prognosis [9,44], the process of cell death has been proposed as the underlying mechanism of ARDS following SARS-CoV-2 infection [11,45]. However, experimental identification and characterization of SARS-CoV-2 infection-triggered programmed cell death are mainly focused on respiratory epithelial cells in rodent models [11,21,22], and the underlying mechanism remains unclear. In this study, we observed extensive apoptotic signals detected in lung tissues of humans and NHPs following SARS-CoV-2 infection. The TUNEL-IF double labeling assay identified cell-type specific apoptosis in NHP lung tissue following SARS-CoV-2 infection on day 21 p.i., mainly in ECs and alveolar type 1 and 2 cells. Both intrinsic and extrinsic apoptotic pathways are activated following SARS-CoV-2 infection. Co-culture models revealed that cell apoptosis can be induced either directly or indirectly by SARS-CoV-2 infection in human cells. Pharmacological activators of EPAC1 protect ECs from apoptosis in Vero cells and HUVECs co-cultures following SARS-CoV-2 infection.

Mounting evidence shows that cell-type apoptosis contributes to the pathogenesis and resolution of ARDS and vasculopathy in the lung [16]. A recent proposal regarding cell apoptosis in the pathogenesis of COVID-19 was based on a histopathological study in vulnerable patients with complicated preexisting endothelial dysfunction, which is associated with multiple factors, including smoking, hypertension, diabetes, obesity, and established cardiovascular diseases. All these diseases are associated with adverse outcomes in COVID-19. Thus, the pathogenesis of cellular apoptosis following SARS-CoV-2 infection and underlying mechanisms remain to be dissected. EC dysfunction has been proposed as a major player in the pathogenesis of COVID-19 [46]. However, to our knowledge, there are not yet published data relevant to endothelial apoptosis in an experimental model of SARS-CoV-2 infection, i.e., NHP [1,18,23,24], hamster [25,26], or mouse models [15,31]. In the present study, we identified cell-type apoptosis occurring in alveolar cells, ECs, macrophages, and T cells following SARS-CoV-2 infection in NHPs.

Cell apoptosis can be induced either directly by the virus itself or indirectly by host inflammatory response during viral infections [11,19,22,47,48]. Many cells undergo apoptosis in response to viral infection as a host defense by reducing the release of progeny virus [41]. However, dysregulation of apoptosis is associated with detrimental effects to the host. Similar to a previous report using a SARS-CoV infection model [32], we observed that primary HUVECs and human lung bronchial BEAS2B cell lines are refractory to SARS-CoV-2 infection when compared with Vero cells. Furthermore, no apoptosis was detected after SARS-CoV-2 challenge. However, our in vivo data demonstrated that apoptosis predominantly takes place in ECs. Thus, we employed a co-culture model to evaluate the potential indirect effect of permissive cells (i.e., Vero cells) on non-permissive cells (i.e., HUVECs or BEAS2S cells) in inducing apoptosis. First, we observed that co-culture did not change the permissiveness of HUVECs or BEAS2B cells and Vero cells to SARS-CoV-2 virus. Interestingly, after co-culture with permissive Vero cells, SARS-CoV-2 infection triggered apoptosis in non-permissive HUVECs or BEAS2B cells at 72 h p.i., supporting our in vivo cell-type apoptosis data regarding different cell lineages, permissive or relatively non-permissive cells. However, the mechanism by which underlying SARS-CoV-2 infection indirectly triggers the apoptosis in relatively non-permissive ECs remains unclear. Veras et al. reported that SARS-CoV-2-triggered neutrophil extracellular traps promote the death of lung epithelial cell lines [19]. Hoagland et al. showed apoptosis in the bronchial epithelium, neutrophil extracellular traps in the bronchioles, and vascular edema in the lungs in a hamster model of SARS-CoV-2 infection [49]. Cytokine-mediated inflammatory cell death has been recently proposed as an underlying mechanisms of programmed cell death triggered by SARS-CoV-2 infection. The combination of TNFα and IFNγ induced apoptosis and other programmed cell deaths in inflammatory cells [11]. Treating with neutralizing antibodies against TNFα and IFNγ reduced the mortality in a mouse model of SARS-CoV-2 infection [11]. Such considerations will provide potential targets for our future studies exploring the signaling pathway(s) that regulate apoptosis in various cell types following SARS-CoV-2 infection.

Apoptosis can be initiated through one of two alternative convergent pathways. The intrinsic pathway is mediated by mitochondria, and the extrinsic pathway is mediated by cell surface death receptors [50,51,52]. Furthermore, perturbations in mitochondria morphology and function in human lung epithelial cells have been documented following SARS-CoV-2 infection [10]. Our data suggest both intrinsic and extrinsic apoptotic pathways are involved in apoptosis following SARS-CoV-2 infections.

We observed significant increases in apoptotic signals in postmortem lung sections from COVID-19 patients and lungs of NHPs infected with SARS-CoV-2 on day 21 p.i., and apoptotic signals were visualized in vascular intima and microvascular ECs. Increasing evidence shows that apoptosis plays a pathogenic role in the chronic phase of virus-induced diseases [53]. Apoptosis of ECs has been associated with increased pro-coagulant properties [54] and enhances the development of atherosclerosis [55]. The potential correlation(s) among SARS-CoV-2-triggered apoptosis in various cell types and vasculopathies, e.g., SARS-CoV-2-associated multisystem inflammatory syndrome in children (MIS-C) [56,57] or long-term conditions such as atherosclerosis [55], remains to be elucidated.

cAMP is one of the most common and universal second messengers. cAMP regulates a myriad of important biological processes under physiological and pathological conditions [58]. As such, several microbial pathogens have evolved a set of diverse virulence-enhancing strategies that exploit the cAMP signaling pathways of their hosts [42,59]. cAMP-based cell signaling mediated by intracellular cAMP receptors EPAC 1 and 2 is one of major contributors to the transduction of the effects of cAMP, and EPAC1 is the major isoform in ECs [60]. The role of EPAC1 in response to viral infection has been demonstrated in MERS and SARS-CoV [59]. EPAC2, not EPAC1, has been proposed as an important factor controlling respiratory syncytial virus replication and virus-induced host responses in human upper and lower airway epithelial cells [61]. In our recent studies, we have shown that inactivation of EPAC1 can attenuate filoviral infection in ECs by blocking the very early stage of infection [42]. We also reported that *EPAC1^–/–^* mice were significantly less susceptible to spotted fever rickettsial invasion compared with wild-type mice [62]. Furthermore, growing evidence has revealed cAMP-EPAC signaling axes play various roles in regulating cell apoptosis in different cells. However, these in vitro data were obtained from cell models utilizing serum starvation due to the use of cAMP analogues as EPAC agonists and cAMP phosphodiesterase in serum [42]. Using non-cyclic nucleotide small molecule EPAC1-specific ESA I942 in normal culture media, we evaluated whether the cAMP-EPAC system participates in SARS-CoV-2 infection-triggered apoptosis. We observed that post-exposure treatment with pharmacological activator of EPAC1 reduces apoptosis of ECs in Vero cells and HUVECs co-cultures following SARS-CoV-2 infection, suggesting that the cAMP-EPAC signaling pathway is a potential target for controlling SARS-CoV-2 infection-induced apoptosis. However, given that the cAMP-EPAC axes are involved in different stages of virus infection [42,59,61], the possible side effect of pharmacological activation of EPAC remains to be addressed.

There were major limitations in our present study. First, NHP data are limited due to the small number of animals and the fact that lung tissue samples were not collected until 21 days post challenge. This significantly restricts our profiling of the course of apoptosis in various cell types following SARS-CoV-2 infection. Further work is needed to characterize the apoptotic events during an active infection at different time points p.i. and to evaluate the potential impact of EPAC1 activator using in vivo models. Second, many cells undergo apoptosis as a host defense to restrict virus amplification [41]. However, dysregulation of apoptosis is associated with pathological resolution following viral infections. Based on the current body of data, the present study fails to determine any potential regulatory roles of apoptosis during the pathogenesis of SARS-CoV-2 infection in permissive cells. Third, a dysregulated innate immune response and endothelial dysfunction have been proposed as playing a role in the pathogenesis of SARS-CoV-2-associated MIS-C [14,63]. The presence of disease-associated autoantibodies against ECs was reported in a COVID-19 patient with MIS-C [64]. Investigation of endothelial programed cell death in multiple organs will help define the pathological mechanism of MIS-C. Lastly, cell-type apoptosis contributes to the resolution of ARDS and vasculopathy in the lung [16]. Long-term deleterious or beneficial consequences of endothelial apoptosis triggered by SARS-CoV-2 infection, which may promote or prevent vasculopathy progression, will be another important research area in the context of the prognosis of COVID-19. Future answers to these questions will shed new light on the potential for a novel therapeutic avenue, wherein the pathogenetic mechanisms underlying apoptosis in various cell types during cytokine storm can be exploited as a treatment for COVID-19.

## 4. Materials and Methods

### 4.1. Clinical Specimens

Procurement of human postmortem tissues for research and all histopathologic procedures were approved by the University of Texas Medical Branch (UTMB; Galveston, TX), Committee on Research with Post-Mortem Specimens. Autopsy procedures were performed under strict CDC guidelines in a negative pressure autopsy suite utilizing battery powered air purifying respirators and proper personal protective equipment [65]. All appropriate postmortem permissions, precautions, and procedures done were according to the guidelines of the College of American Pathologists [65].

### 4.2. Biosafety Level and Study Subjects

Two male and one female rhesus macaques (age: 46–48 months, weight: 3–5 kg, Envigo, Alice, TX, USA) and two female cynomolgus macaques (age: 84–112 months, 3–5 kg, Envigo) were used in a COVID-19 model development study. One of the cynomolgus macaques was used as an unchallenged control. The formalin-fixed specimens were donated for retrospective histopathological assessment in the present study. The NHPs were challenged with 5.0 × 10^8^ TCID_50_ of SARS-CoV-2 (isolate 2019-nCoV/USA/WA1/2020) via the intranasal route (0.5 mL/naris) using the MAD Nasal™ Intranasal Mucosal Atomization Device) and the intratracheal (4.0 mL) routes. The virus was propagated in Vero E6 cells. Viral titer (TCID_50_) was determined on Vero E6 cells [22]. The NHPs were monitored for 21 days post-challenge and euthanized. Organs were collected in 10% buffered formalin. We also used archived normal NHP tissue lung sections (from one rhesus macaque and one cynomolgus macaque) purchased from Zyagen (San Diego, CA, USA) as controls.

### 4.3. Co-Culture of Vero Cells and HUVECs or Vero Cells and BEAS2B Cells

For co-culturing of Vero cells and HUVECs or Vero cells and BEAS2B cells, Vero cells were seeded in the inserts of 24-well plates (0.4 µm polyester membrane, Costar, Thermo Fisher Scientific) while HUVECs or BEAS2B cells were grown in wells without inserts. After 48 h, the insert with Vero cells were transferred to the well of HUVECs or BEAS2B cells. Cells were then infected by adding SARS-CoV-2 at a multiplicity of infection (MOI) of 0.1 to the inserts. The co-cultures were kept in co-culture media (HUVECs and Vero cells co-culture medium was ½ HUVEC medium and ½ Vero cell medium; Vero cells and BEAS2B cells co-culture medium was Vero cell medium). After 72 h, media were removed and cells were fixed with 10% buffered formalin. For the EPAC agonist or antagonist treatment, the co-culture media for Vero cells and HUVECs were replaced with 5 µM I942 or NY0173 [42]-supplemented media at 24 h p.i. At 72 h p.i., media were removed and cells were fixed with 10% buffered formalin. Dimethyl sulfoxide (DMSO)-supplemented medium was used as the vehicle. All experiments were performed in replicates of three or more.

### 4.4. In Situ TUNEL Staining and Quantitative Analysis of TUNEL Signals

For the fluorescein terminal deoxynucleotidyl transferase dUTP nick end labeling (TUNEL) assay, we followed the manufacturer’s protocol [11,22]. All samples were thoroughly rinsed with PBS for 5 min three time before the assay. After antigen retrieval and blocking, slides were rinsed twice with PBS and excess fluid drained. The TUNEL reaction mixture was added and incubated with labeling solution for 60 min at 37 °C, protected from light. Negative controls were incubated with labeling solution only. Nuclei were stained with DAPI. Fluorescent images were reviewed using an Olympus BX51-microscope prior to further analysis using a Nikon A1R MP ECLIPSE T*i* confocal microscope, with *NIS*-Elements imaging software version 4.50.00 (Nikon, Tokyo, Japan). A published protocol was used to do quantitative analysis of the extent of TUNEL signal-positive cells in the co-cultures of Vero cells and HUVECs. Briefly, TUNEL and DAPI signals were captured with an Olympus BX51 image system using a final 10× optical zoom. The extent of TUNEL signal-positive cells is normalized to total nuclear content (DAPI staining) in each filed using NIH ImageJ [66]. Twenty-six microscopic fields were examined for each sample. Data are representative of at least three experiments.

### 4.5. IF-TUNEL Dual Staining

For IF-TUNEL dual staining, after IF using CD31, caveolin-1, SPC, CD68, CD3, cytochrome C, FasL, or SARS-CoV-2 antibody, the TUNEL assay was done following the protocol above. Fluorescent images were reviewed using an Olympus BX51-microscope prior to further analysis using a Nikon A1R MP ECLIPSE T*i* confocal microscope.

### 4.6. Statistical Analysis

Values are reported as mean ± SEM. The statistical significance was determined using Student’s *t*-test or one-way ANOVA analysis. Statistical significance was considered as *p* < 0.05.

## Figures and Tables

**Figure 1 pathogens-10-00509-f001:**
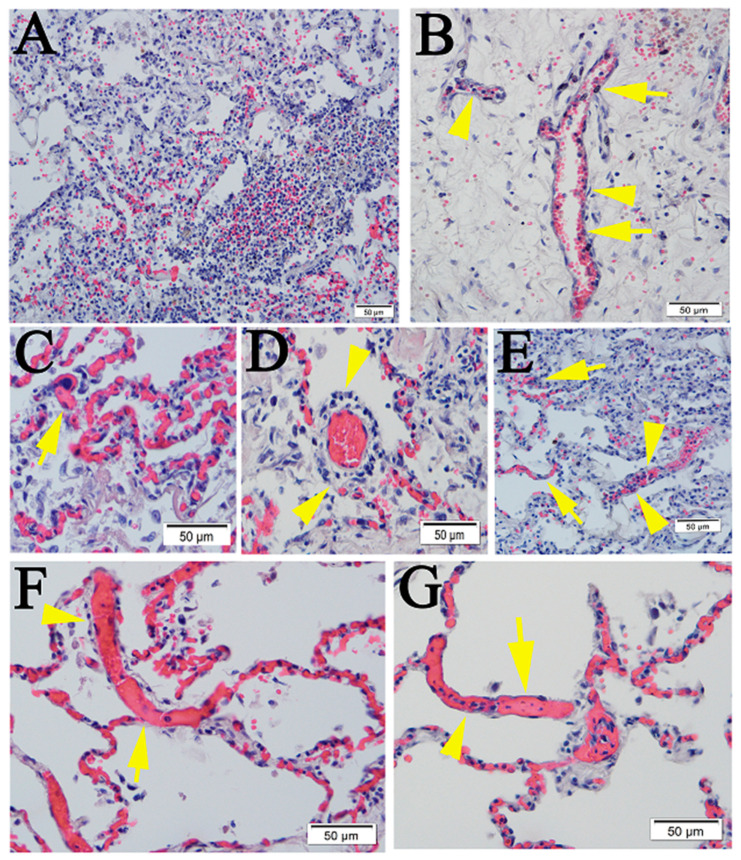
Pulmonary pathology in human cases COVID-19. (**A**) Area of lymphocytic pneumonia. (**B**) Dominant lymphocyte (arrow heads) and monocyte (arrows) adhesion and infiltration within the intima along the lumen of blood vessels. (**C**) Microthrombus (arrow) and extensive endotheliitis. (**D**) Lymphocytic perivasculitis (arrow heads). (**E**) Alveolar septae widened by inflammatory cells (arrow heads) and lymphocytic endotheliitis (arrows). (**F**,**G**) Extensive microthrombi in the capillaries of the septae (arrows) and dilated vessels as opposed to the nearby congested capillaries (arrow heads). Compare with normal human lung (Appendix A from an archival paraffin tissue block). Scalebars, 50 µm.

**Figure 2 pathogens-10-00509-f002:**
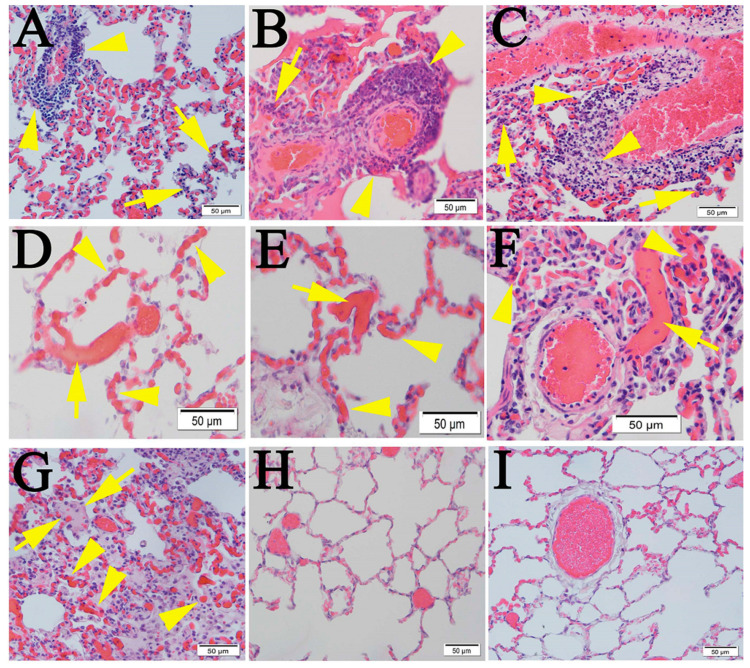
Pulmonary pathological changes in NHPs infected with SARS-CoV-2. Three rhesus macaques and one cynomolgus macaque were intranasally and intratracheally challenged with 5 × 10^8^ TCID50 of SARS-CoV-2 (isolate 2019-nCoV/USA/WA1/2020). Lung tissues were harvested on day 21 p.i. Lung tissues from one normal cynomolgus macaque and lung sections from two normal NHPs (obtained commercially) were employed as mock controls. (**A**–**C**) Monocyte perivascularitis (arrow heads) and endotheliitis (arrows). (**D**–**F**) Extensive microthrombi (arrows) and congestions (arrow heads) in the capillaries of the septae. (**G**) Intra-alveolar edema (arrows), heavily congested capillaries (arrow heads), and septal inflammation. (**H**) Uninvolved regions of the lung in the same animal, compared with uninfected mock control (**I**). Scalebars, 50 µm.

**Figure 3 pathogens-10-00509-f003:**
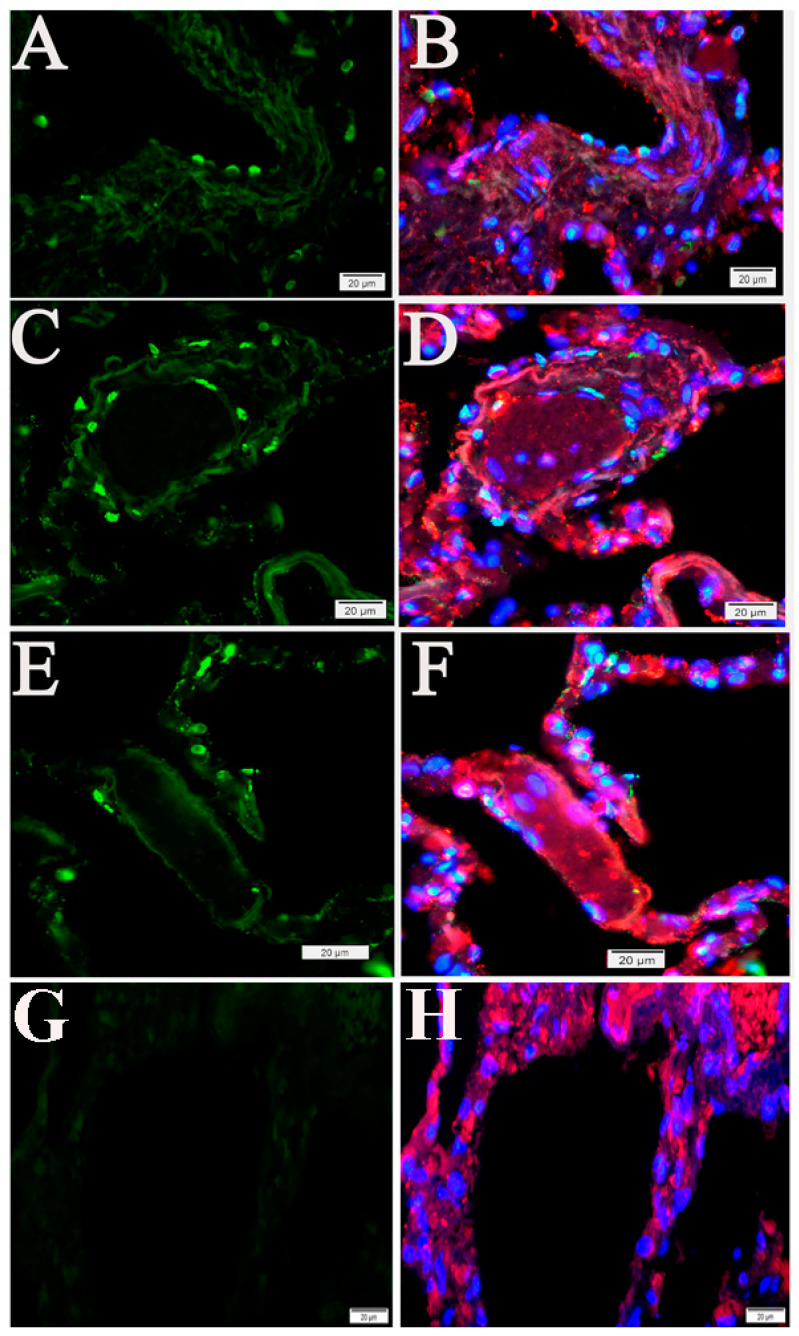
Apoptosis in human postmortem lung sections. (**A**–**F**) Tissue from COVID-19 patients. (**G**,**H**) Normal human lung. Representative TUNEL-IF double labeling demonstrated apoptosis (green) and EC-specific CD31 (red) in postmortem lung sections from COVID-19 patients with different underlying diseases. Nuclei of human cells were counterstained with DAPI (blue). Scalebars, 20 µm.

**Figure 4 pathogens-10-00509-f004:**
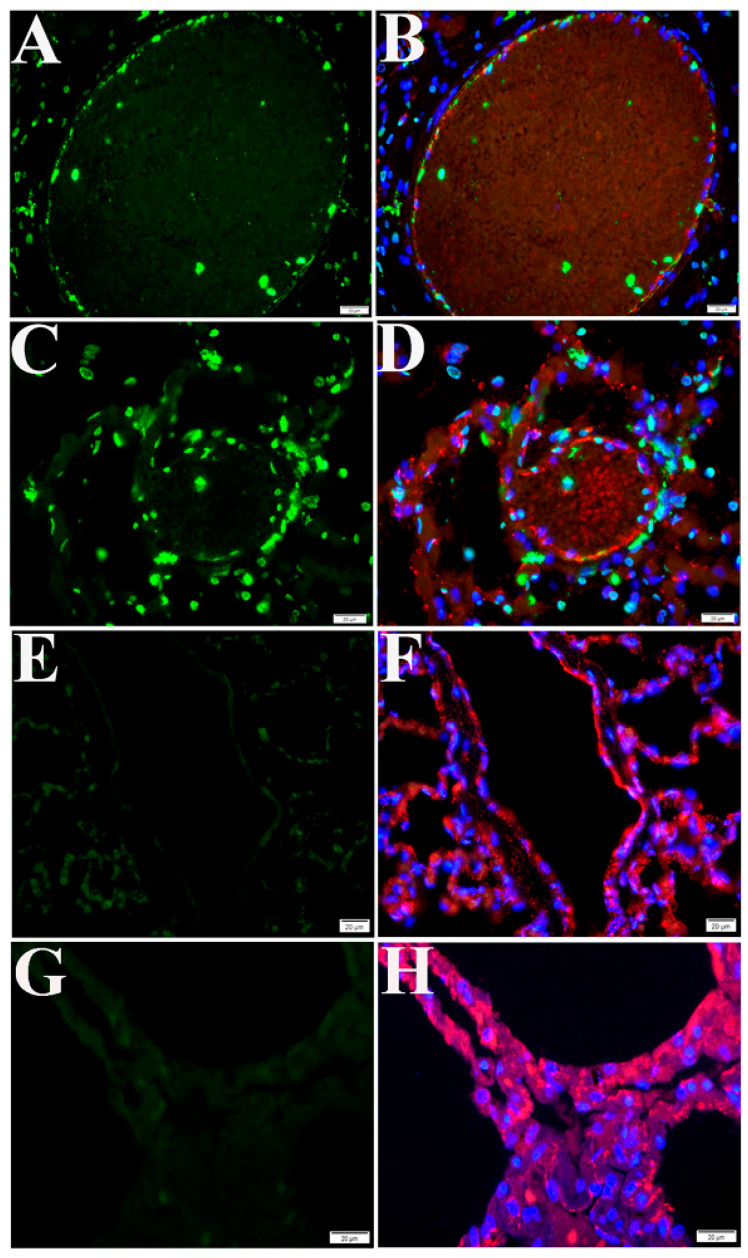
Apoptosis in lung tissue of NHPs. (**A**–**D**) Lung tissue on day 21 p.i. with SARS-CoV-2. (**E**,**F**) Uninvolved regions of the lungs in the same animal. (**G**,**H**) Uninfected mock control tissue. NHP lung tissues were harvested on day 21 p.i. Representative TUNEL-IF double labeling demonstrated apoptosis (green) and EC-specific CD31 (red) in SARS-CoV-2-infected NHP lung tissue. Nuclei of NHP cells were counterstained with DAPI (blue). Scalebars, 20 µm.

**Figure 5 pathogens-10-00509-f005:**
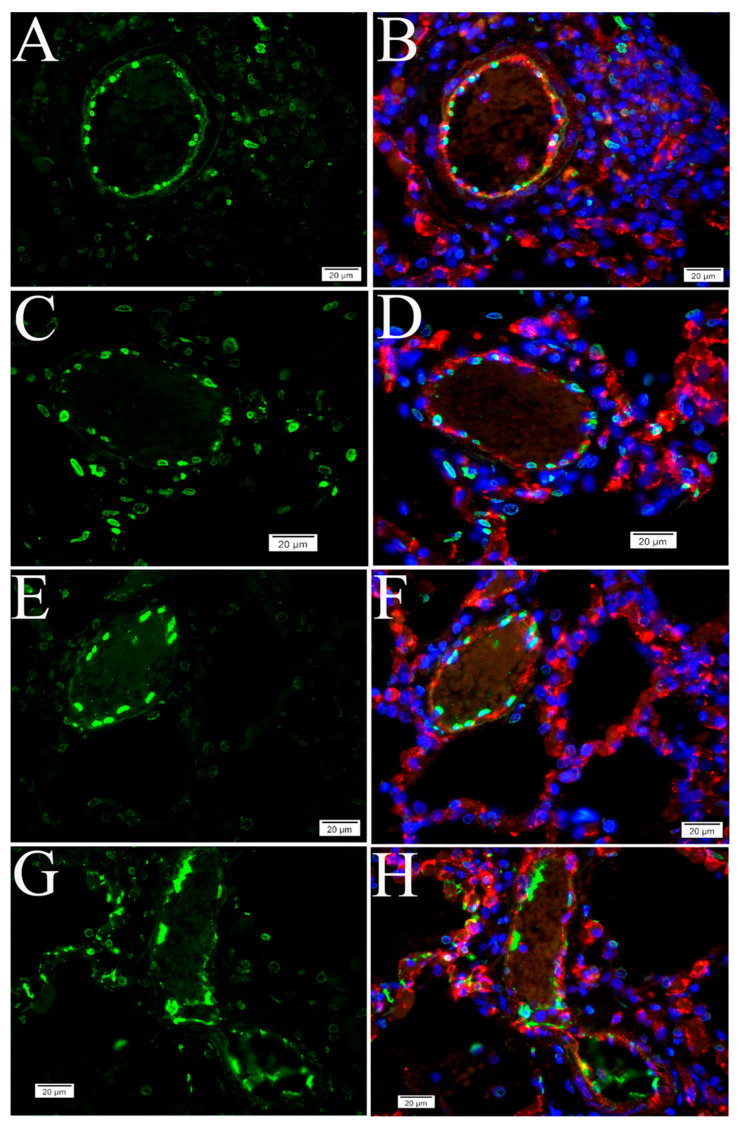
Apoptosis occurs in vascular intima and microvascular ECs. (**A**–**H**) TUNEL signals (green) detected in CD31(red)-marked ECs in blood vessels and capillaries in SARS-CoV-2 infected NHP lung tissue, compared with the controls in Figure 4G,H. Nuclei of NHP cells were counterstained with DAPI (blue). Scalebars, 20 µm.

**Figure 6 pathogens-10-00509-f006:**
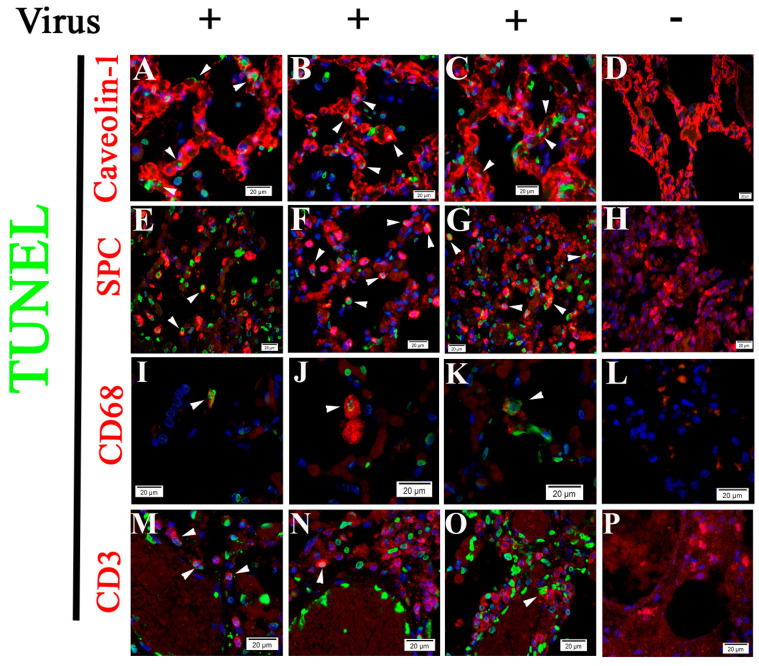
Apoptosis occurs in alveolar type 1 (**A**–**C**) and 2 cells (**E**–**G**), macrophages (**I**–**K**), and T cells (**M**–**O**). TUNEL signals (green) (arrow heads) detected in both caveolin-1 (red)-marked (**A**–**C)** and SPC (red)-marked **(E**–**G**) cells in alveoli of SARS-CoV-2 infected NHP lung tissue. TUNEL signals (green) (arrow heads) were detected in CD68 (red)-marked macrophages (**I**–**K**) and CD3 (red)-marked T cells (**M**–**O**) in SARS-CoV-2 infected NHP lung tissue. (**D**,**H**,**L**,**P**) Control tissue. Nuclei of NHP cells were counterstained with DAPI (blue). Scalebars, 20 µm. Apoptotic signals were visualized in caveolin-1-(23.99 ± 1.64%), SPC-(17.44 ± 1.86%), CD68-(1.18 ± 0.22%), CD3-(16.08 ± 2.39% positive cells in infected groups *n* = 4). No apoptotic signal was detected in uninfected control group (*n* = 3). One field under fluorescent microscope (20×) was captured in one stained tissue section per case and image was processed with Image J to calculate the ratio of both TUNEL- and caveolin-1-, SPC-, CD68-, or CD3-positive cells to total TUNEL cells.

**Figure 7 pathogens-10-00509-f007:**
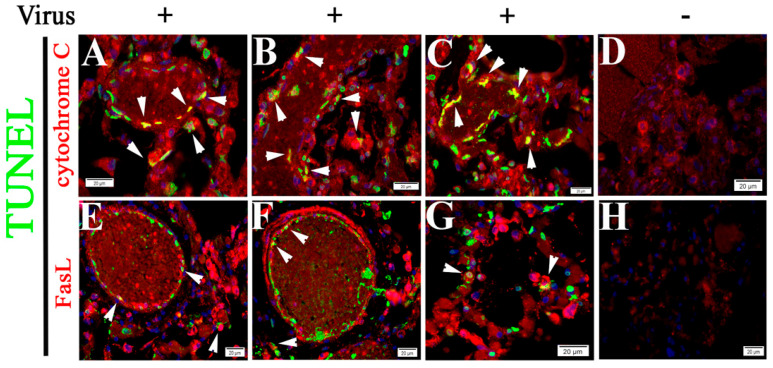
The intrinsic and extrinsic apoptotic pathways are activated in NHP lung tissue during SARS-CoV-2 infection. (**A**–**C**) Colocalizations (arrow heads) between TUNEL signals (green) and IF signals of cytochrome C (red) were visualized in alveoli of SARS-CoV-2 infected NHP lung tissue. (**D**) Mock control tissue. (**E**–**G**) Colocalizations (arrow heads) between TUNEL signals (green) and IF signals of FasL (red) were visualized in alveoli of SARS-CoV-2 infected NHP lung tissue. (**H**) Mock controls. Nuclei of NHP cells were counterstained with DAPI (blue). Scalebars, 20 µm.

**Figure 8 pathogens-10-00509-f008:**
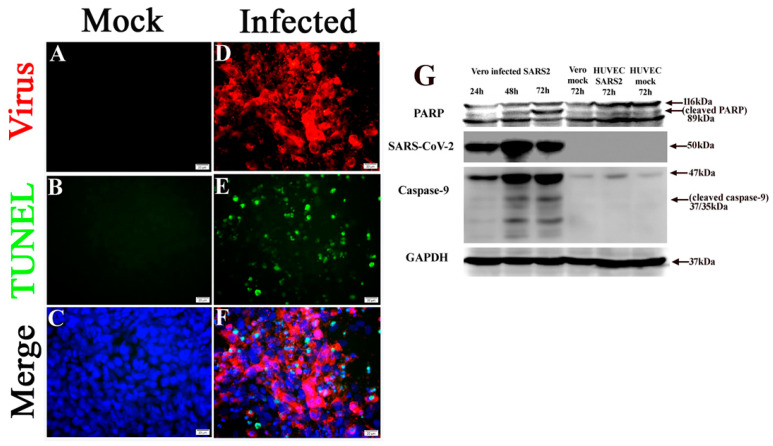
SARS-CoV-2 infection-induced apoptosis in Vero cells. (**A**–**C**) Uninfected controls. (**D**–**F**) TUNEL signals (green) detected in Vero cells at 72 h p.i. with 0.1 MOI SARS-CoV-2 (red). Nuclei of cells counterstained with DAPI (blue). Scalebars, 20 µm. (**G**) Western blotting shows SARS-CoV-2 infection triggered cleavages of PARP and caspase-9 at 72 h p.i. in Vero cells but not in HUVECs.

**Figure 9 pathogens-10-00509-f009:**
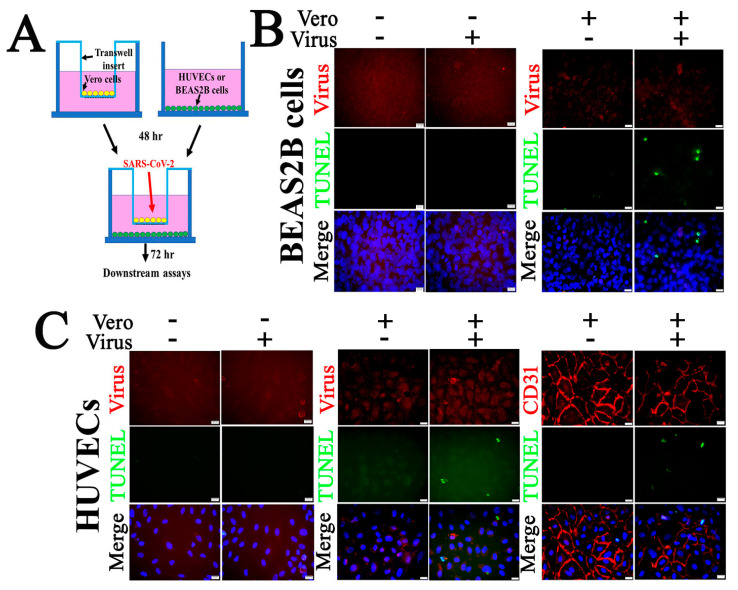
Co-culture of Vero cells and relatively nonpermissive HUVECs or BEAS2B cells results in apoptosis following SARS-CoV-2 infection. (**A**) Co-culture of Vero cells and BEAS2B (**B**) or HUVECs (**C**) cells. HUVECs (**C**) or BEAS2B cells (**B**) (in the culture wells) were co-cultured with or without Vero cells (in the culture insert). The co-cultures were exposed to 0.1 MOI of SARS-CoV-2 for 72 h. Fixed PBS thoroughly rinsed HUVECs (**C**) or BEAS2B cells (**B**) in the well were subjected to IF staining to detect SARS-CoV-2 (red) or CD31(**C**) (red; an EC-specific marker) and TUNEL (green). Fixed Vero cells in the insert were subjected to IF staining to detect SARS-CoV-2 and TUNEL (Appendix A). Nuclei of HUVECs were counterstained with DAPI (blue). Scalebars, 20 µm. Apoptotic signals were visualized in BEAS2B cells (11.23 ± 1.12%) and HUVECs (12.44 ± 2.24%) in infected groups (*n* = 9/group). No apoptotic signal was detected in uninfected control groups (*n* = 9/group). Three fields under fluorescent microscope (20×) were captured per well and image was processed with Image J to calculate the ratio of TUNEL to DAPI-labeled nucleated cells.

**Figure 10 pathogens-10-00509-f010:**
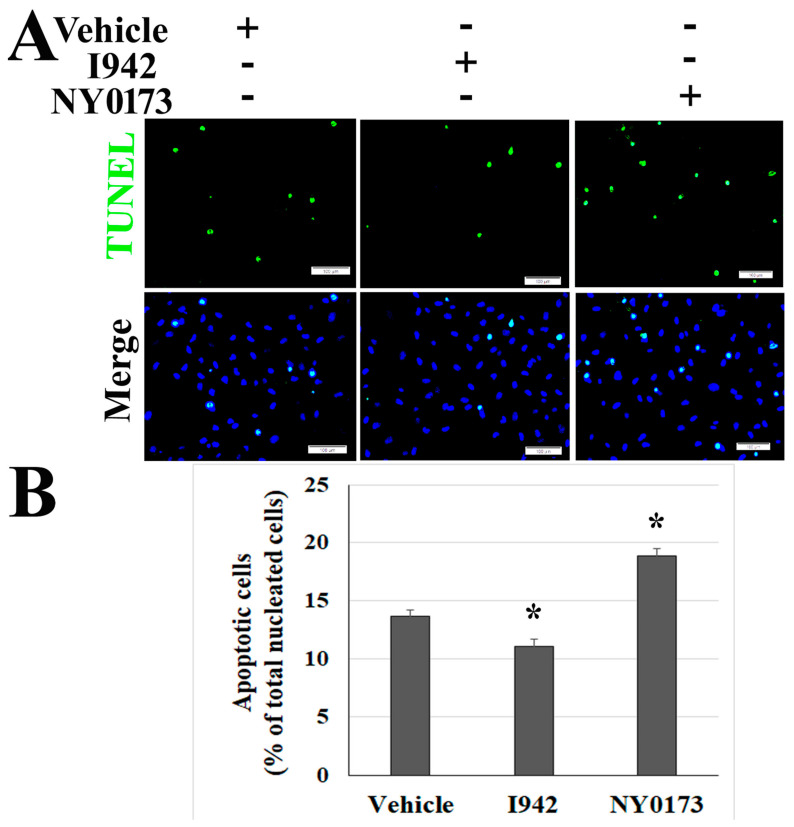
Pharmacological activators of EPAC protect ECs from apoptosis in Vero cells and HUVECs co-cultures following SARS-CoV-2 infection. HUVECs were co-cultured with Vero cells. The co-cultures were exposed to 0.1 MOI of SARS-CoV-2 for 24 h before treatment with I942 (5 µM), NY0173 (5µM), and vehicle for 48 h. Fixed HUVECs in the well were subjected to IF staining for SARS-CoV-2 (red) (Appendix A) and TUNEL assay (green) (**A**). Fixed Vero cells in the inserts were subjected to TUNEL assay (Appendix A). Nuclei of HUVECs were counterstained with DAPI (blue). Scale bars, 100 µm. (**B**) Percentage of TUNEL signal-positive HUVECs among all nucleated cells (DAPI staining) in each filed. The quantitative data presented are three independent experiments (*n* = 78 fields in each group). Values are reported as mean ± SEM. Statistical significance was determined using a one-way analysis of variance. * compared with the vehicle group, *p* < 0.01.

## Data Availability

No new data were created or analyzed in this study. Data sharing is not applicable to this article.

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
