# Peer review of "Cell-Type Apoptosis in Lung during SARS-CoV-2 Infection"

_pathogens, 2021, doi:10.3390/pathogens10050509_

Round 1
Reviewer 1 Report
Many thanks for your manuscript. It has been a pleasure to read and comment on it.
Besides, I have some comments that could polish the merits of your manuscript as follows:
1-Please, use another word than ameliorated in the abstract because it could be confusing. You perhaps can use reduced apoptosis to keep the meaning clear for the reader and not confused until you reach the discussion part?
2- Would you please change the keyword "TUNEL" with another keyword because it could be misleading or ambiguous for the reader.
3- In the Introduction, when you mentioned atypical pneumonia, please, open that to express why it's atypical and please dive deeper into the mentioned associated cytokine storm.
4- Please, re-write some sentences to avoid identically mentioning them in different parts (e.g., has inspired renewed interest in understanding) in the abstract and introduction.
5- In the discussion, would you please argue in-depth about the differences between the direct viral cell apoptosis and the indirect apoptosis induced by the host-inflammatory response?
6- Please, remove the italics in the discussion, if it is intentional, to let the readers come to the important take-away points after reading your argumentation.
7- I am afraid that the collection of lung samples after 21 days could affect the soundness of the conclusions. Would it be possible that the authors perform another small experiment to collect the lung samples in a shorter period of time after the challenge and argue about that?
8- Please, correct in the discussion SARS-CoV-19 into SARS-CoV-2.
9- Please, add an extra section for the conclusions and future directions to highlight the most important findings of your study and recommend areas for further investigations (e.g., future studies on the mechanisms underlying SARS-CoV-2 indirect apoptosis).
Author Response
#1 Reviewer
Many thanks for your manuscript. It has been a pleasure to read and comment on it.
Besides, I have some comments that could polish the merits of your manuscript as follows:
1-Please, use another word than ameliorated in the abstract because it could be confusing. You perhaps can use reduced apoptosis to keep the meaning clear for the reader and not confused until you reach the discussion part?
The reviewer’s comment is correct. We have replaced the “ameliorated” word with “reduced”.
2- Would you please change the keyword "TUNEL" with another keyword because it could be misleading or ambiguous for the reader.
We have revised it to “TUNEL assay”.
3- In the Introduction, when you mentioned atypical pneumonia, please, open that to express why it's atypical and please dive deeper into the mentioned associated cytokine storm.
The reviewer’s comment is correct. We have replaced with a more appropriate term “COVID-19 pneumonia” and cite relevant references. We have also added more information in the Discussion regarding the cytokine storm.
4- Please, re-write some sentences to avoid identically mentioning them in different parts (e.g., has inspired renewed interest in understanding) in the abstract and introduction.
This sentence has been revised.
5- In the discussion, would you please argue in-depth about the differences between the direct viral cell apoptosis and the indirect apoptosis induced by the host-inflammatory response?
The reviewer’s comment is correct. Our response is similar to that of comment #3.
6- Please, remove the italics in the discussion, if it is intentional, to let the readers come to the important take-away points after reading your argumentation.
We have corrected it.
7- I am afraid that the collection of lung samples after 21 days could affect the soundness of the conclusions. Would it be possible that the authors perform another small experiment to collect the lung samples in a shorter period of time after the challenge and argue about that?
As we claimed, tissue samples were from retrospective studies in NHP and clinical samples, restricting us to expand to more time points. However, this comment is critical for our future study and we have revised relevant information in the Discussion about future directions.
8- Please, correct in the discussion SARS-CoV-19 into SARS-CoV-2.
We have corrected this error.
9- Please, add an extra section for the conclusions and future directions to highlight the most important findings of your study and recommend areas for further investigations (e.g., future studies on the mechanisms underlying SARS-CoV-2 indirect apoptosis).
We agree with the reviewer. We have revised the last paragraph of the Discursion as the conclusion and future directions.
Reviewer 2 Report
This is a well written manuscript desribing relatedness between different type of the infected infected cells with SARS CoV 2 and apoptosis induction. The topic of this paper is onr of the highest importance to public health and life.
Author Response
#2 Reviewer
This is a well written manuscript desribing relatedness between different type of the infected infected cells with SARS CoV 2 and apoptosis induction. The topic of this paper is one of the highest importance to public health and life.
We would like to thank you for evaluating our manuscript.
Reviewer 3 Report
Liu, et al. aims to characterize the extent of apoptotic events in SARS-CoV-2 infections and their impact in the development and severity of COVID-19, using post-mortem COVID-19 patient samples, NHP and cell culture. Their main conclusions are that apoptosis is a major component of the pathogenesis associated to SARS-CoV-2 infection, that the induction of apoptosis involves both intrinsic and extrinsic pathways, that non-permissible cells can still undergo apoptosis mediated by secreted molecules (cytokines), and finally suggest that targeting the cAMP-EPAC signaling pathway could be a potential therapeutic to ameliorate the detrimental effects of infection-induced apoptosis. Although the study clearly shows that apoptosis is a frequent occurrence in SARS-CoV-2 infections, there is an overall lack of statistic rigor in the experiments, and a poor organization and visualization of the results.
Major issues
1) Lack of statistical rigor and poor organization of results: the authors perform extensive imaging experiments to describe apoptosis and other histopathological consequences of infection, as well as to characterize the different cell types capable of undergoing apoptosis. However, the data presented could have much more impact if besides a representative image of the apoptotic events, there is a quantitative measure of these events (% of apoptotic cells, % of overlap between TUNEL signal and IF, etc.) and perform proper statistics (n=5 for patients and n=4 for NHP and include more controls). This is particularly necessary for Fig 6 and Fig 7, since from these images it is not clear the extent by which macrophages and t-cells undergo apoptosis, neither the relative frequency of the activation of extrinsic and intrinsic apoptotic pathways. Besides, it is not clear if the images correspond to different sections of the same animal/patient sample or instead different patients (in particular because the authors show on average 3 images but they have 5 patients and 4 animals, and use insufficient controls). Additionally, all figures need a better organization to include proper and clear controls (not include them in a separate supplemental figure), and add labels for clarity (of the different colors and conditions). Since the effects are subtle, Fig9 would also benefit to be presented in a quantitative way (% of apoptotic cells in each condition). Finally, for figure 10B, the authors should expand on the nature of the quantification (is the % of apoptotic cells calculated for 78 cells in each condition for one of three independent experiments? if this is the case where the other 2 experiments also significant?) and what are the statistics (what does the error bars represent, and what test was used to calculate significance).
2) As the authors mention, the cAMP-EPAC signaling is involved in “a myriad of important biological processes”. Therefore, the way the manuscript is structured, it is not clear why the authors decided to target EPAC1 to study apoptosis, especially given the several other molecules historically associated to apoptosis. Furthermore, while the authors data shows a reduction or induction of apoptotic events using drugs that target EPAC1, it is not clear what effect this will have in the control of SARS-CoV-2 infection, nor in its pathogenesis. One suggestion would be to test the impact of their EPAC1 activator in animal models of infection (hamsters, ACE2-mice, NHP), and address the consequences of inhibiting apoptosis, although I understand that this point might be out of scope for this manuscript (but the authors could at least include a discussion about this).
3) As the authors mention, apoptosis is usually considered as an antiviral mechanism, but it could also contribute to pathogenesis if not properly regulated. Therefore, the authors should expand their discussion to include the nature of the secreted molecule responsible for the apoptosis of non-infected cells and discuss the impact that inhibiting virus-mediated apoptosis in general (as a therapy) would have to disease progression (side-effects), and in particular given that the cAMP-EPAC signaling is involved in several different biological processes. Furthermore, Hoagland et al. (https://doi.org/10.1016/j.immuni.2021.01.017) shows apoptosis in the bronchial epithelium as well as nested accumulations of neutrophils in the bronchioles of infected hamsters, so they should be referenced.
Minor issues
- Define ARDS in the abstract.
- Citation missed at the end of the first paragraph of page 3.
- The supplemental table should include further information related to the infection of the patient (days after hospitalization, days after symptom onset, etc.).
- Reference to Fig.S1 in page 3 should be Fig. S1G-I)
- Fig1: add control image for comparison, and clearly state the number of controls used.
- Legend of Fig2D-F: both microthrombi and congestions are mentioned as arrows.
- FigS2: the mock is loaded at insufficient amounts that it makes hard to compare the DNA fragmentation. Are these samples from fixed-tissues? any concern on DNA fragmentation from fixed tissue?
- Fig3H and 4H lack DAPI stain.
- Given Figure 4, is there a purpose for Fig5?
- Fig 6 and 7 need labels and its hard to follow. As mentioned previously this data would have much more impact if it can be expressed in a quantitative way.
- Legend of Fig7: “Nuclei of NHP cells were counterstained with DAPI” is repeated twice.
- Fig8: add labels to cell images.
- Fig9: is there a reason why there is high background signal of the virus (red) in Vero co-cultured cells? Also move Fig9C to Fig9A for clarity.
- Reference 32 is not the best to describe this sentence.
- Figure 10A: I don’t understand what is the difference between panel Fig10A and FigS4D? is that another replicate of the experiment? Why not showing the virus panel corresponding to Fig10A, instead of a completely different experiment?
- Is there any reason why EPAC1 targeting drugs had no effect in Vero cells (the cells actually infected with SARS-CoV-2?). Maybe the apoptosis caused by the virus is different from the one caused by the secreted signaling molecule.
- Figure 10 legend: the authors should refrain from combining the description of supplemental figures here.
- End of page 17: “SARS-CoV-19-associated”
- The authors should include in methods a small description of the growth conditions for their SARS-CoV-2 stocks.
Author Response
#3 Reviewer
Liu, et al. aims to characterize the extent of apoptotic events in SARS-CoV-2 infections and their impact in the development and severity of COVID-19, using post-mortem COVID-19 patient samples, NHP and cell culture. Their main conclusions are that apoptosis is a major component of the pathogenesis associated to SARS-CoV-2 infection, that the induction of apoptosis involves both intrinsic and extrinsic pathways, that non-permissible cells can still undergo apoptosis mediated by secreted molecules (cytokines), and finally suggest that targeting the cAMP-EPAC signaling pathway could be a potential therapeutic to ameliorate the detrimental effects of infection-induced apoptosis. Although the study clearly shows that apoptosis is a frequent occurrence in SARS-CoV-2 infections, there is an overall lack of statistic rigor in the experiments, and a poor organization and visualization of the results.
Major issues
1) Lack of statistical rigor and poor organization of results: the authors perform extensive imaging experiments to describe apoptosis and other histopathological consequences of infection, as well as to characterize the different cell types capable of undergoing apoptosis. However, the data presented could have much more impact if besides a representative image of the apoptotic events, there is a quantitative measure of these events (% of apoptotic cells, % of overlap between TUNEL signal and IF, etc.) and perform proper statistics (n=5 for patients and n=4 for NHP and include more controls). This is particularly necessary for Fig 6 and Fig 7, since from these images it is not clear the extent by which macrophages and t-cells undergo apoptosis, neither the relative frequency of the activation of extrinsic and intrinsic apoptotic pathways. Besides, it is not clear if the images correspond to different sections of the same animal/patient sample or instead different patients (in particular because the authors show on average 3 images but they have 5 patients and 4 animals, and use insufficient controls). Additionally, all figures need a better organization to include proper and clear controls (not include them in a separate supplemental figure), and add labels for clarity (of the different colors and conditions). Since the effects are subtle, Fig9 would also benefit to be presented in a quantitative way (% of apoptotic cells in each condition). Finally, for figure 10B, the authors should expand on the nature of the quantification (is the % of apoptotic cells calculated for 78 cells in each condition for one of three independent experiments? if this is the case where the other 2 experiments also significant?) and what are the statistics (what does the error bars represent, and what test was used to calculate significance).
These comments are important.
Quantitative analysis of image data: The relative frequency of the activation of extrinsic and intrinsic apoptotic pathways has been assessed using ImageJ and added in the result 2.4 and Supplemental Table 2.
Control images: During basic histology study we employed the well-published strategy in designing and presenting H&E staining on control and experimental samples (Rockx B, et al, Science 2020;368:1012) (Munster VJ, et al, Nature 2020;382:692). Multiple images from infected tissues are presented in the manuscript to show different pattern of the pathology in the lung. However, single image is presented to show well-known histology of normal lung tissue and included in Supplemental Materials. Thus, there is enough space in the figures to present image data in as detail as possible. More information and references have been added in Supplementary Methods section.
Different color labels: Information regarding different color signals has been added by different labels in revised Figures 6, 7, and 8.
The nature of the quantification in Fig. 10: The error has been corrected with “ 78 fields” in the legend.
2) As the authors mention, the cAMP-EPAC signaling is involved in “a myriad of important biological processes”. Therefore, the way the manuscript is structured, it is not clear why the authors decided to target EPAC1 to study apoptosis, especially given the several other molecules historically associated to apoptosis. Furthermore, while the authors data shows a reduction or induction of apoptotic events using drugs that target EPAC1, it is not clear what effect this will have in the control of SARS-CoV-2 infection, nor in its pathogenesis. One suggestion would be to test the impact of their EPAC1 activator in animal models of infection (hamsters, ACE2-mice, NHP), and address the consequences of inhibiting apoptosis, although I understand that this point might be out of scope for this manuscript (but the authors could at least include a discussion about this).
These comments are important. We cited a wrong reference (previous 32). Now we cited a new reference, about “Differential roles of EPAC in regulating cell death in neuronal and myocardial cells”, supporting our test the potential role of EPAC in apoptosis in SARS-CoV-2 infection. More information has been added in the future directions in Discussion.
3) As the authors mention, apoptosis is usually considered as an antiviral mechanism, but it could also contribute to pathogenesis if not properly regulated. Therefore, the authors should expand their discussion to include the nature of the secreted molecule responsible for the apoptosis of non-infected cells and discuss the impact that inhibiting virus-mediated apoptosis in general (as a therapy) would have to disease progression (side-effects), and in particular given that the cAMP-EPAC signaling is involved in several different biological processes. Furthermore, Hoagland et al. (https://doi.org/10.1016/j.immuni.2021.01.017) shows apoptosis in the bronchial epithelium as well as nested accumulations of neutrophils in the bronchioles of infected hamsters, so they should be referenced.
These comments are important, combined with the comments from Reviewer #1. We have added new information in Discussion regarding the direct viral cell apoptosis and the indirect apoptosis induced by the host-inflammatory response. The new reference has been cited.
Minor issues
- Define ARDS in the abstract.
We have added the definition of ARDS in the abstract.
- Citation missed at the end of the first paragraph of page 3.
A citation has been added.
- The supplemental table should include further information related to the infection of the patient (days after hospitalization, days after symptom onset, etc.).
Information about the days from symptom onset to death has been added in the Note of the Table.
- Reference to Fig.S1 in page 3 should be Fig. S1G-I)
Yes, we have revised it.
- Fig1: add control image for comparison, and clearly state the number of controls used.
Information has been added in the legends of Fig. 1 and Fig. S1.
- Legend of Fig2D-F: both microthrombi and congestions are mentioned as arrows.
The reviewer’s comment is correct, we have corrected this error.
- FigS2: the mock is loaded at insufficient amounts that it makes hard to compare the DNA fragmentation. Are these samples from fixed-tissues? any concern on DNA fragmentation from fixed tissue?
As we described in the Method of DNA extraction, DNA samples were extracted from formalin-fixed NHP lung tissues using the DNeasy Blood & Tissue kit, following manufacturer’s instruction in particular for formalin-fixed tissue samples. We have revised this part further.
- Fig3H and 4H lack DAPI stain.
We have corrected this error.
- Given Figure 4, is there a purpose for Fig5?
Fig. 4 is generally about Apoptosis in lung tissue, using vascular endothelial marker to visualize the blood vessel as a anatomical mark. Fig. 5 is about endothelial cell type.
- Fig 6 and 7 need labels and its hard to follow. As mentioned previously this data would have much more impact if it can be expressed in a quantitative way.
We have revised Fig. 6 and Fig. 7 by adding relevant labels. The quantitative data have been added in Supplemental Table 2.
- Legend of Fig7: “Nuclei of NHP cells were counterstained with DAPI” is repeated twice.
Yes, we have corrected this error.
- Fig8: add labels to cell images.
Labels have been added in the new Fig. 8.
- Fig9: is there a reason why there is high background signal of the virus (red) in Vero co-cultured cells? Also move Fig9C to Fig9A for clarity.
Currently, we don’t know the reason about the background.
- Reference 32 is not the best to describe this sentence.
We have corrected this error by replacing with correct reference 34.
- Figure 10A: I don’t understand what is the difference between panel Fig10A and FigS4D? is that another replicate of the experiment? Why not showing the virus panel corresponding to Fig10A, instead of a completely different experiment?
Fig. 10A is for showing the representative images during quantification. Fig. S4D is for showing more detail information under higher power objective of fluorescent microscope.
- Is there any reason why EPAC1 targeting drugs had no effect in Vero cells (the cells actually infected with SARS-CoV-2?). Maybe the apoptosis caused by the virus is different from the one caused by the secreted signaling molecule.
Currently we don’t know the scientific reason.
- Figure 10 legend: the authors should refrain from combining the description of supplemental figures here.
Similar to the comment above, Fig. 10A is for showing the representative images during quantification. Fig. S4D is for showing more detail information under higher power objective of fluorescent microscope.
- End of page 17: “SARS-CoV-19-associated”
We have corrected this error.
- The authors should include in methods a small description of the growth conditions for their SARS-CoV-2 stocks.
Relevant information has been added under 4.2.
Round 2
Reviewer 1 Report
I could not find the added conclusion section. It will be appreciated if the authors highlight their conclusions in a separate section, as they have mentioned in their reply!
Author Response
Added information in the last paragraph in the Discussion has been colorfully highlighted as additional information for the conclusion.
Reviewer 3 Report
While the manuscript was certainly improved by the addition of labels to figures, there are a few comments that were not properly addressed:
1) Quantitative information for data on figure 6: Just from the images it is not clear the extent by which macrophages and t-cells undergo apoptosis (% of specific cells that undergo apoptosis).
2) Since the effects are subtle, Fig9 would also benefit to be presented in a quantitative way (% of apoptotic cells in each condition).
3) Fig10B: The data presented is said to be “representative of 3 independent experiments” Can the authors include in the supplemental data the graphs corresponding to the other 2 experiments (showing statistical significance)
4) While the authors mention that future studies are needed to test the impact of EPAC1-targeted drugs, the authors should at least speculate the possible side-effects of inhibiting virus-mediated apoptosis as a therapy and to disease progression. Specially given that the cAMP-EPAC signaling is involved in “a myriad of important biological processes”.
Minor issues:
1) Page 17 line 4: missing citation
Author Response
Reviewer #3
While the manuscript was certainly improved by the addition of labels to figures, there are a few comments that were not properly addressed:
1) Quantitative information for data on figure 6: Just from the images it is not clear the extent by which macrophages and t-cells undergo apoptosis (% of specific cells that undergo apoptosis).
In Figure 6 legend, we have added the quantitative data.
2) Since the effects are subtle, Fig9 would also benefit to be presented in a quantitative way (% of apoptotic cells in each condition).
In Figure 9 legend, we have added the quantitative data.
3) Fig10B: The data presented is said to be “representative of 3 independent experiments” Can the authors include in the supplemental data the graphs corresponding to the other 2 experiments (showing statistical significance)
The reviewer is correct. The inappropriate using “representative of “ in the legend of Figure 10 makes reader confused. We have corrected this error by removing this term. So do we correct the similar error in the legend of supplemental Fig.S4.
4) While the authors mention that future studies are needed to test the impact of EPAC1-targeted drugs, the authors should at least speculate the possible side-effects of inhibiting virus-mediated apoptosis as a therapy and to disease progression. Specially given that the cAMP-EPAC signaling is involved in “a myriad of important biological processes”.
This comment is important. Relevant information has been added in the Discussion (page 17, the end of paragraph 4).
Minor issues:
1) Page 17 line 4: missing citation
Reference 11 has been cited here.